# A parametric framework for multidimensional linear measurement error regression

Stanley Luck *

Vector Analytics LLC, Wilmington, DE, United States of America

* stan.luck@vectoranalytics.ai

## Abstract

The ordinary linear regression method is limited to bivariate data because it is based on the Cartesian representation $y = f(x)$. Using the chain rule, we transform the method to the parametric representation $(x(t), y(t))$ and obtain a linear regression framework in which the weighted average is used as a parameter for a multivariate linear relation for a set of linearly related variable vectors (LRVVs). We confirm the proposed approach by a Monte Carlo simulation, where the minimum coefficient of variation for error (CVE) provides the optimal weights when forming a weighted average of LRVVs. Then, we describe a parametric linear regression (PLR) algorithm in which the Moore-Penrose pseudoinverse is used to estimate measurement error regression (MER) parameters individually for the given variable vectors. We demonstrate that MER parameters from the PLR and nonlinear ODRPACK methods are quite similar for a wide range of reliability ratios, but ODRPACK is formulated only for bivariate data. We identify scale invariant quantities for the PLR and weighted orthogonal regression (WOR) methods and their correspondences with the partitioned residual effects between the variable vectors. Thus, the specification of an error model for the data is essential for MER and we discuss the use of Monte Carlo methods for estimating the distributions and confidence intervals for MER slope and correlation coefficient. We distinguish between elementary covariance for the $y = f(x)$ representation and covariance vector for the $(x(t), y(t))$ representation. We also discuss the multivariate generalization of the Pearson correlation as the contraction between Cartesian polyad alignment tensors for the LRVVs and weighted average. Finally, we demonstrate the use of multidimensional PLR in estimating the MER parameters for replicate RNA-Seq data and quadratic regression for estimating the parameters of the conical dispersion of read count data about the MER line.

## 1 Introduction

In this work, we consider the problem of fitting a multidimensional line for data that are subject to stochastic error. The motivation for this work comes from a collaborative R & D effort involving the application of the genome-wide association (GWAS) [1] and eQTL methods [2] to identify beneficial agronomic variation in maize. This led to our applied algebraic investigation of the merits of various effect size measures and their associated statistical methodologies

**Data Availability Statement:** The RNA-Seq GLDS-251 dataset is available from the GeneLab Omics database at https://genelab-data.ndc.nasa.gov/genelab/accession/GLDS-251/. The textbook, "Statistical Inference" by G. Casella and R. L. Berger, also serves as a data source.

**Funding:** The author, Stanley Luck, is a member of Vector Analytics LLC, which is a science consulting company. The funder provided support in the form of salaries for authors [SL], but did not have any additional role in the study design, data collection and analysis, decision to publish, or preparation of the manuscript. The specific roles of these authors are articulated in the 'author contributions' section.

**Competing interests:** There are no competing interests connected with our consulting work at Vector Analytics LLC. This affiliation also does not alter our adherence to PLOS ONE policies on sharing data and materials. This work is not associated with any patents or commercial products.

as described in our two recent publications in this journal. In the first, we discussed the importance of factoring a $2 \times 2$ contingency table for obtaining marginal scale invariant effect size measures for proportional variation [3]. We also used projective geometric concepts for discussing connections between the phi coefficient, odds ratio, relative risk and proportion with respect to the representation of effect sizes. In the second, we discussed the formulation of a nonparametric measure for nonoverlap proportion and the importance of specifying a complete set of parameters for point-biserial variation in the formulation of an effect size measure [4]. We also discussed the importance of providing an error model for data and the use of Monte Carlo (MC) methods for estimating distributions and confidence intervals for effect size, as required for best practices [5]. However, in RNA-Seq studies, error estimation is complicated by the low number of replications usually employed. Our RNA-Seq experimental design is consistent with the recommended minimum of three replicates [6], but this is less than the minimum of 13 replicates required for reliably estimating variance [7]. Furthermore, RNA-Seq read counts can range over more than three orders of magnitude with heteroscedastic error. Thus, weighted least squares (WLS) methods [8, 9] are needed for properly partitioning residual effects in data analyses. The importance of accounting for measurement errors in omics studies has been discussed in several publications [7, 10–13]. These problems provide the motivation for our investigation of linear measurement error regression (MER) methods for multivariate data. The concepts discussed in this paper significantly extend our previous gene expression data analysis work that is briefly described in [14].

We use the term 'measurement error regression' in accordance with the recommendation in [15] instead of the other commonly used terms, such as 'error-in-variables' or 'major axis regression' [16]. There are many studies on the MER problem [17–19], but [9, 20–22] were particularly useful in this work. There are two forms of WLS optimization in measurement error regression for $(\mathbf{x}, \mathbf{y})$ data. The elementary form involves the application of the normal equations [9] algorithm to obtain WLS-based linear regression (WR) estimates for the necessary parameters. However, WR is subject to the limitation where a design matrix is constructed from error free independent data, and the residual effects are assigned exclusively to the dependent data. Thus, WR requires the specification of 'dependent' and 'independent' quantities. Then, when both $\mathbf{x}$ and $\mathbf{y}$ are subject to error the WR estimate is biased, and there is extensive literature describing various approaches for solving this problem [18–20]. The second form of WLS involves 'weighted orthogonal regression' (WOR) or 'orthogonal distance regression' where the weighted sum of squares of the combined residual effects for $\mathbf{x}$ and $\mathbf{y}$ are minimized [23]. The 'dependent' versus 'independent' distinction for $\mathbf{y}$ and $\mathbf{x}$ does not hold in WOR. Then, WOR can be regarded as a generalization because under the condition where only the $\mathbf{y}$ data are subject to error, the WOR and WR estimates are equivalent. However, there is no analytical solution for WOR [23], and the implementation requires a numerical iterative search method. In this work, we use the implementation of WOR in open source statistics software [24, 25] based on the nonlinear ODRPACK algorithm [26]. However, ODRPACK is limited to the analysis of bivariate data because it is based on the Cartesian representation $y = f(x)$. Our objective is to describe a multivariate generalization of the WR algorithm that provides unbiased estimates for MER parameters for a set of linearly related variable vectors (LRVVs). The main novel contributions of this work are as follows: 1) We associate a set of LRVVs with a convex set of weighted averages and use Monte Carlo simulation to demonstrate that the minimum coefficient of variation for error (CVE) provides the optimal weights for forming the weighted average (Eqs 13, 16 and 17). 2) We formulate a parametric linear regression (PLR) algorithm where the weighted average of the LRVVs serves as the independent parameter, and the MER parameters are estimated using the Moore-Penrose pseudoinverse (Eqs 22 and 23; Proposition 1). 3) Convex mappings from the data points

in the MER graph to the predicted values along the MER line are provided (Eqs 6 and 7). 4) We propose that in the parametric representation for the linear relation between variable vectors, the corresponding covariance is a parametric vector quantity (Eq 25; Proposition 2). 5) We identify scale invariant quantities for the partitioned residual effects between the LRVVs for both PLR (Eq 27) and WOR (Eq 37). 6) We propose a multivariate generalization of the Pearson correlation coefficient based on the $m$-fold contraction of Cartesian polyad alignment tensors for the LRVVs and weighted average (Eqs 32, 33 and 35). 7) A quadratic regression method is used to estimate the conical dispersion parameters for determining the read counting error in replicated RNA-Seq data (Eqs 40 and 43).

## 2 Methods

Section 2.1 introduces our notations. In sections 2.2–2.4, we examine the WLS methods for partitioning residual effects in linear regression for $(\mathbf{x}, \mathbf{y})$ data. In section 2.5, we discuss the minimum CVE criterion for obtaining the optimal weighted average of a set of LRVVs. In section 2.6, we describe the PLR framework where parametric equations are used to obtain a novel multidimensional generalization of the normal WLS algorithm for linear regression. In section 3.2, we describe the application of the PLR algorithm for the analysis of the conical dispersion of replicated RNA-Seq data about the MER line.

It has already been established that the specification of an error model $\mathcal{E}$ for the given data is essential in MER analysis. An error model serves as a realistic assessment of the performance of a data acquisition system, and error parameters are estimated from replicated measurements [27]. The parameters are usually summarized as an 'expected variance' for error with respect to each observation. When referring to experimental error, we use the term 'expected variance' because error parameters are empirical. We provide a detailed discussion of the role of $\mathcal{E}$ in partitioning residual effects for the WLS optimization of MER. Furthermore, proper statistical practice dictates that effect size measures such as the regression slope, the correlation coefficient and Cohen's $d$ must be qualified by a distribution [5, 28], and an assessment of substantive significance [29] must account for a distributed system response. Consequently, we also discuss the use of Monte Carlo methods and $\mathcal{E}$ for estimating the distributions and confidence intervals of MER parameters.

### 2.1 Notation

Our notation and terminology are partly taken from [30], but comprehensive discussions of vector spaces and linear algebra for applied statistics are found in many textbooks, such as [8, 9, 31]. Scalars are denoted by italics $y$, and vectors are written in bold lowercase letters $\mathbf{y} \equiv (y_i) \equiv (y_1, y_2, y_3, \ldots, y_{\dim(\mathbf{y})})$. Matrices are written in bold uppercase letters $\mathbf{Y} \equiv [y_{ij}]$, and the transpose is denoted as $\mathbf{Y}^T$. Then, $n$ joint observations for $m$ experimental quantities produce a dataset $\{y_{ij} \mid y_{ij} \in \mathbb{R}^1, 1 \le i \le n, 1 \le j \le m\}$, which corresponds to the Cartesian product of the data vectors $\mathcal{Y} = (\mathbf{y}_j), \mathbf{y}_j \in \mathbb{R}^n$, where $\dim(\mathcal{Y}) = n \times m$. We use a convenient terminology for the subsets of $\mathcal{Y}$, where each axis of a regression graph is assigned to a *variable vector* $\mathbf{y}_j$, the linear combinations $\Sigma_j k_j \mathbf{y}_j$ are elements of the *variable space*, and each point in the graph corresponds to an *observation vector* $\mathbf{y}_{(i)}$ [30, 32]. In this work, we describe a novel parametric linear regression algorithm for estimating MER parameters for a set $\mathcal{Y}$ of LRVVs. We also use the shortened term 'variable vectors' synonymously when referring to LRVVs, except in a few clearly identified cases. The inner product with the one-vector $\mathbf{1} = (1, 1, 1, \ldots)$ produces the mean value $\bar{y} = \mathbf{1} \cdot \mathbf{y}/\dim(\mathbf{y})$. The subtraction of the mean vector produces a centered vector $\mathbf{y}_c = \mathbf{y} - \bar{y}\mathbf{1}$. The hat symbol '^' denotes a unit-length vector, i.e., $\parallel \hat{\mathbf{y}} \parallel = 1$. Then, the Pearson correlation coefficient $r(\mathbf{x}, \mathbf{y})$ is defined as the cosine of the angle between unit-length, centered

vectors [30, 32]

$$r \quad = \hat{\mathbf{x}}_c \cdot \hat{\mathbf{y}}_c. \tag{1}$$

The variance for $\mathbf{y}_j$ is denoted as $\text{Var}(\mathbf{y}_j)$. The expected variance for normally distributed errors $e_{y_{ij}}$ is denoted as $s_{y_{ij}}^2$, and the vector $\mathbf{s}_{\mathbf{y}_j}^2$ denotes the expected variance of the error in $\mathbf{y}_j$. For bivariate data $(\mathbf{y}_j, \mathbf{y}_k)$, the covariance is denoted as $\text{Cov}(\mathbf{y}_j, \mathbf{y}_k)$; the notation for a variance-covariance matrix is not needed in this work. In our $\text{Cov}(\mathbf{y}_j, \mathbf{y}_k)$ notation, $\mathbf{y}_j$ and $\mathbf{y}_k$ are independent and dependent quantities, respectively, and the roles are reversed for $\text{Cov}(\mathbf{y}_k, \mathbf{y}_j)$. The connection between correlation and covariance as measures of linear dependence in data is discussed in [21, 32]. A general discussion of convex sets, linear fractional transformations and perspective functions is found in [33]. The set containing all convex combinations of a set of variable vectors $\mathcal{Y}$ is denoted as $\mathbf{conv}(\mathcal{Y})$, where $\tau \in \mathbf{conv}(\mathcal{Y}) \Rightarrow \tau = \sum_j w_j \mathbf{y}_j$, for $\Sigma_j\, w_j = 1$ and $0 \le w_j \forall j$. A perspective function has the form $P(\mathbf{y}, t) = \mathbf{y}/t$. In recent publications, we provide an elementary discussion about projective geometry for fractional variation [3], and identify the homogeneous coordinates of the Pearson correlation coefficient as points on the line passing through the point $(r, \sqrt{1 - r^2}) \in \mathbb{S}_+^1$ [4]. Equivalent homogeneous coordinates are indicated by a two-sided arrow, i.e., $\mathbf{y} \leftrightarrow \mathbf{x} \Rightarrow \mathbf{y} = \mathbf{x}\, t$.

## 2.2 The WLS linear regression algorithm

Consider a set of $n$ linearly related observations $\{(x_i, y_i)|1 \le i \le n\}$, which are represented as the variable vectors $\{(\mathbf{x}, \mathbf{y})|\mathbf{x}, \mathbf{y} \in \mathbb{R}^n\}$. The observations are subject to error, and the differences between the true $(\mathbf{x}^*, \mathbf{y}^*)$ and observed values are denoted by $(\mathbf{e_x}, \mathbf{e_y}) = (\mathbf{x} - \mathbf{x}^*, \mathbf{y} - \mathbf{y}^*)$. The fact that $(\mathbf{x}^*, \mathbf{y}^*)$ are unknown implies that $(\mathbf{e_x}, \mathbf{e_y})$ are also indeterminate. The specification of an error model for the data $\mathcal{E}(\mathbf{x}, \mathbf{y})$ is essential for estimating the distributions and corresponding confidence intervals of the statistical parameters, including the slope; see section 2.6. The default approach is to reduce $\mathcal{E}(\mathbf{x}, \mathbf{y})$ to a stochastic form or some approximation thereof, where the errors $(\mathbf{e_x}, \mathbf{e_y})$ are independent and associated with normal distributions with zero means $(\mathbf{0}, \mathbf{0})$ and expected variances $(\mathbf{s_x}^2, \mathbf{s_y}^2)$, where $\{\text{Cov}(e_{x_i}, e_{x_k}) = 0 \mid i \ne k\}$, $\{\text{Cov}(e_{y_i}, e_{y_k}) = 0 \mid i \ne k\}$ and $\{\text{Cov}(e_{x_i}, e_{y_k}) = 0 \,\forall\, i, k\}$. In the standard MER framework [20, 21], the linear relation between $\mathbf{x}$ and $\mathbf{y}$ is expressed using Cartesian equations:

$$\mathbf{y} \quad = \alpha_{xy}\mathbf{1} + \beta_{xy}\mathbf{x}^* + \mathbf{e_y}, \tag{2}$$

$$= \alpha_{xy}\mathbf{1} + \beta_{xy}\mathbf{x} + (\mathbf{e_y} - \beta_{xy}\mathbf{e_x}), \tag{3}$$

where $\{\alpha_{xy}, \beta_{xy} \in \mathbb{R}^1\}$ are the model parameters. We use the subscript '$xy$' to indicate the correspondence with the derivative $\beta_{xy} = dy_i^*/dx_i^*$, because it will be necessary to distinguish between different parameterizations in Eq 21. In linear regression analysis, the objective is to obtain sample estimates $(a_{xy}, b_{xy})$ for $(\alpha_{xy}, \beta_{xy})$. After introducing the design matrix $\mathbf{X} = [\mathbf{1}, \mathbf{x}]$ and the diagonal variance matrix $\mathbf{V_y} = \text{diag}(\mathbf{s_y}^2)$, the WLS linear regression estimate is obtained by solving the normal equations to obtain [9]

$$\mathbf{u}_{xy,\text{WR}} \quad = (\mathbf{X}^T \mathbf{V_y}^{-1} \mathbf{X})^{-1} \mathbf{X}^T \mathbf{V_y}^{-1} \mathbf{y}, \tag{4}$$

where $\boldsymbol{u}_{xy,\text{WR}} = (a_{xy,\text{WR}}, b_{xy,\text{WR}})$. $\mathbf{V_y}$ is required for the WLS scaling of both $\mathbf{X}$ and $\mathbf{y}$, and $(\mathbf{X}^T\mathbf{X})^{-1}\mathbf{X}^T$ is the Moore-Penrose pseudoinverse of $\mathbf{X}$; in practice, the pseudoinverse is estimated by singular value decomposition. The ordinary linear regression (OLR) estimate is obtained when $\mathbf{V_y} \propto \mathbf{I}$, where $\mathbf{I}$ is the identity matrix. However, the $b_{xy,\text{WR}}$ estimate is biased because of

the correlation of $\mathbf{x}$ with $\mathbf{e_x}$ in Eq 3 [20]. The reliability ratio [17] for $\mathbf{x}$ is defined as

$$\kappa_{\mathbf{x}} = 1 - \frac{\mathrm{Var}(\mathbf{e_x})}{\mathrm{Var}(\mathbf{x})},$$

where $\mathrm{Var}(\mathbf{x}) = \mathrm{Var}(\mathbf{x}^*) + \mathrm{Var}(\mathbf{e_x})$, and $\mathrm{Var}(\mathbf{e_x}) = \mathbf{1} \cdot \mathbf{s_x}^2/\dim(\mathbf{s_x}^2)$. As illustrated in Fig 1A and 1B, $b_{xy,\mathrm{WR}}$ is attenuated as $\kappa_{\mathbf{x}}$ decreases from 1. This bias in $b_{xy,\mathrm{WR}}$ arises from the algebraic requirement that $\mathbf{X}$ must serve as an error-free quantity in the normal equations, and the residuals are assigned exclusively to $\mathbf{y}$. The solution $\mathbf{u}_{xy,\mathrm{WR}}$ holds under the condition where $\mathrm{Var}(\mathbf{e_x})$ is small enough such that $\mathbf{x} \approx \mathbf{x}^*$ with $\kappa_{\mathbf{x}} \approx 1$. A key objective of this work is to describe a novel

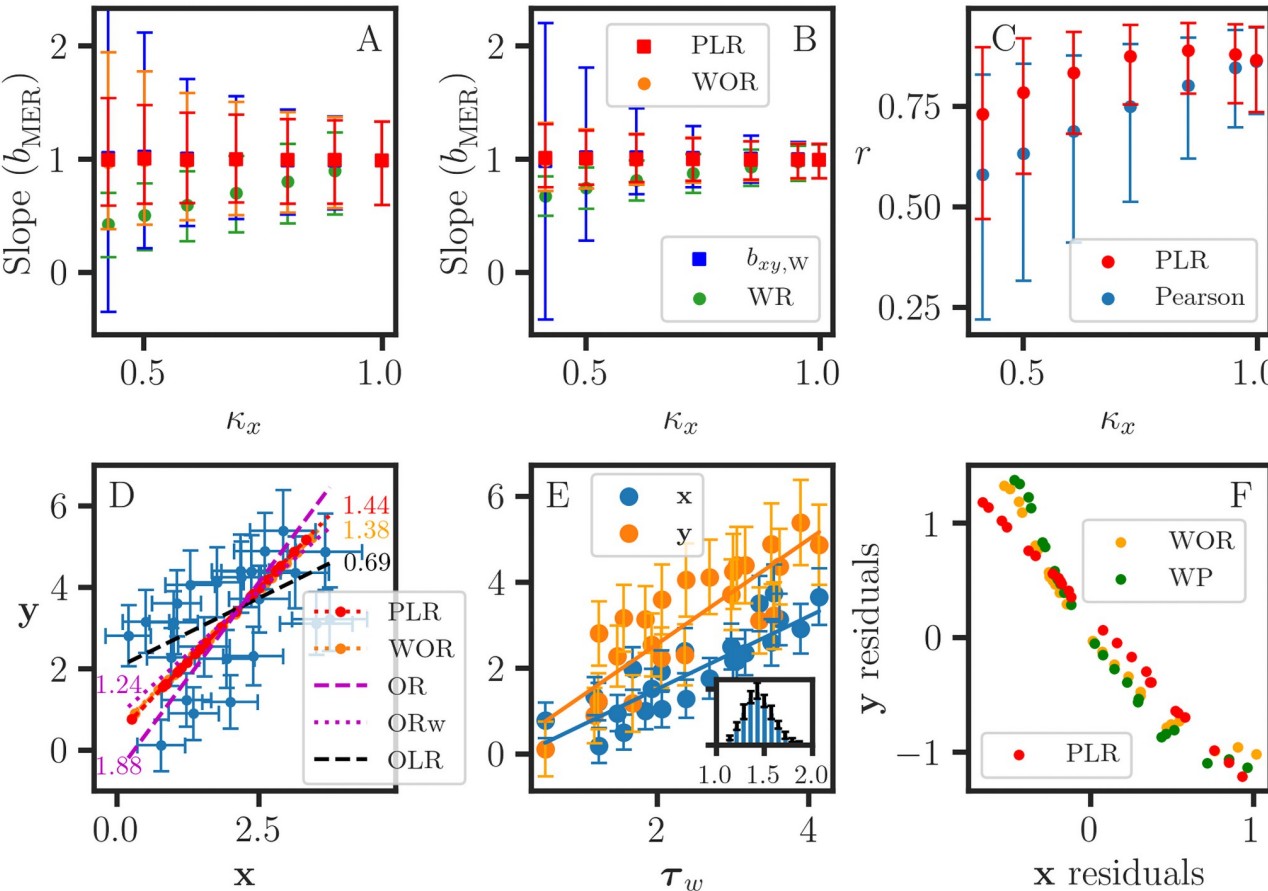

**Fig 1. Comparison of various measurement error regression methods.** A, B, C: Monte Carlo simulations of MER for 25 evenly-spaced values $y_i^* = x_i^*$. The statistical parameters are estimated from 5000 Monte Carlo datasets $(\mathbf{x}, \mathbf{y}) = (\mathbf{x}^* + \mathbf{e_x}, \mathbf{y}^* + \mathbf{e_y})$, with normally distributed errors $(\mathbf{e_x}, \mathbf{e_y})$. The parametric linear regression and orthogonal distance regression methods both produce consistent estimates for the slope ($b_{\mathrm{MER}}$) for a wide range of reliability ratios $\kappa_x$. The weighted average ($b_{xy,\mathrm{W}}$) of the attenuation-corrected direct and inverse WR estimates also produces comparable results. In contrast, the standard ($b_{\mathrm{WR}}$) estimate is attenuated. A: Homoscedastic error: ($\mathbf{s}_{\mathbf{y}^*} = 0.3$, $\mathbf{s}_{\mathbf{x}^*} = \alpha$), and $\alpha = \{0.01, 0.1, 0.15, 0.2, 0.25, 0.3, 0.35\}$. B: Heteroscedastic error: ($\mathbf{s}_{\mathbf{y}^*} = 0.01 + 0.3\mathbf{y}^*$, $\mathbf{s}_{\mathbf{x}^*} = 0.01 + \alpha\mathbf{x}^*$), and $\alpha = \{0.01, 0.1, 0.2, 0.3, 0.4, 0.5, 0.6\}$. C: Using the results from (B), the Pearson correlation coefficient $r_{\mathbf{xy}}$ is attenuated compared to the parametric correlation coefficient $r_{\mathrm{PLR}} = r_{\tau_w \mathbf{x}} r_{\tau_w \mathbf{y}}$ for the weighted average $\tau_w = w\,\mathbf{x} + (1-w)\mathbf{y}$. D, E, F: MER results for the data $(\mathbf{x}, \mathbf{y})$ from Table 11.3.1 in [21]. However, error estimates were not provided for the data. For demonstration purposes, we used a rudimentary error model: $\mathbf{s}_{\mathbf{x}}^2 = 0.04 + 0.005\mathbf{x}^2$, $\mathbf{s}_{\mathbf{y}}^2 = 0.1 + 0.005\mathbf{y}^2$, $\kappa_x = 0.93$, $\kappa_y = 0.91$, $r = 0.49$, and $r_{\mathrm{PLR}} = 0.74$. D: Each MER line is labeled with its slope. The ordinary least squares ($b_{\mathrm{OLR}}$) and orthogonal regression ($b_{\mathrm{OR}}$) estimates are consistent with [21]. The WLS orthogonal regression ($b_{\mathrm{ORw}}$) estimate is also shown. The predicted values for $(x_i, y_i)$ are plotted for PLR and WOR. E: The $b_{xy,\mathrm{PLR}}$ estimate is obtained from separate regressions $b_{\mathbf{rx},\mathrm{WR}}$ and $b_{\mathbf{ry},\mathrm{WR}}$. The inset shows the distribution of $b_{xy,\mathrm{PLR}}$ estimated from 100 histograms with 5000 MC samples per histogram (see Fig 4); the error bars indicate the degree of convergence for the MC simulation. F: The weighted average prediction residuals for $b_{xy} = 1.38$ and the WOR residuals are quite similar. The PLR residuals from the $\tau$ WR results in (E) are comparable but do not account for heteroscedastic error. A-E: The error bars correspond to $\pm 2\sigma$.

parametric generalization in which the WR algorithm itself serves as the basis for measurement error regression for a set of LRVVs (section 2.6). We also compare the performance of PLR with that of existing MER methods and provide realistic MER examples using publicly available data. In particular, we discuss the WLS orthogonal regression and attenuation-corrected slope estimation methods.

## 2.3 WLS orthogonal regression

The WOR algorithm minimizes the weighted sum of squares for the combined residual error for $(e_{x_i}^2, e_{y_i}^2)$:

$$\min_{a,b} \sum_i (w_{y_i}(a + bx_i^* - y_i)^2 + w_{x_i}e_{x_i}^2),$$ (5)

with $(w_{x_i}, w_{y_i}) \propto (1/s_{x_i}^2, 1/s_{y_i}^2)$ [26] and does not require the identification of dependent and independent variables. However, this minimization problem does not possess an analytical solution [23], and a numerical method is needed for the implementation of WOR. For our demonstration, the python scipy.odr package [25] provides a convenient WOR implementation based on the ODRPACK optimization library [26]. We are interested in WOR for a linear function $y = a + bx$, but ODRPACK is more general and uses the Levenberg-Marquardt nonlinear least squares algorithm to support parameter estimation for a user-specified bivariate function $y = f(x)$. As demonstrated in Fig 1, the ODRPACK algorithm performs well for homoscedastic (Fig 1A) and heteroscedastic (Fig 1B) error models over a wide range of reliability ratios. When either $\kappa_x$ or $\kappa_y$ approaches 1, the WOR estimate is equivalent to the WR estimate. Thus, the WOR algorithm is regarded as a generalization of WR. Furthermore, the ODRPACK algorithm provides predicted and residual values for $(x_i, y_i)$, as demonstrated in Fig 1F. For comparison, we describe a novel algebraic method to estimate predicted and residual values in MER. First, we associate the observed data $(x_i, y_i)$ with a convex mapping to an interval on an MER line with (intercept, slope)$\equiv(a, b)$. The interval is bounded by the following two points:

$$\mathbf{v}_{x_i} = (x_i, bx_i + a)$$

and

$$\mathbf{v}_{y_i} = \left(\frac{y_i - a}{b}, y_i\right),$$

which correspond to the special cases $s_{x_i}^2 = 0$ and $s_{y_i}^2 = 0$, respectively. Then, the weighted average

$$\mathbf{v}_{i,w} = w\mathbf{v}_{x_i} + (1 - w)\mathbf{v}_{y_i}$$ (6)

is contained in $\mathbf{conv}(\mathbf{v}_{x_i}, \mathbf{v}_{y_i})$ for $0 \le w \le 1$. Using the signal-to-noise ratios $(\bar{x}/s_{x_i}^2, \bar{y}/s_{y_i}^2)$ (see Eq 17), the optimal weight is obtained:

$$w = \left(1 + \frac{\bar{y}}{\bar{x}} \frac{s_{x_i}^2}{s_{y_i}^2}\right)^{-1},$$ (7)

which produces the weighted average prediction (WP) value $\mathbf{v}_{i,\mathrm{WP}}$. In Fig 1F, we demonstrate that the ODRPACK and WP estimates for residuals are quite similar. The generalization of Eqs 6 and 7 for MER in $\mathbb{R}^m$ is straightforward.

However, the WR and WOR algorithms are both subject to the algebraic limitation that the Cartesian representation $y = f(x)$ only allows for the representation of lines (or curves) in two

dimensions. Alternatively, the specification of a straight line in $\mathbb{R}^m$ requires the use of the parametric representation

$$\mathbf{u} = \boldsymbol{\alpha} + \boldsymbol{\beta} t, \tag{8}$$

with $\mathbf{u}, \boldsymbol{\alpha}, \boldsymbol{\beta} \in \mathbb{R}^m$ and $t \in \mathbb{R}^1$ [34]. Consequently, our objective is to describe a parametric framework where the WR algorithm serves as the basis for measurement error regression for LRVVs.

## 2.4 Covariance in ordinary linear regression

The OLR algorithm corresponds to the special case where $\mathbf{V_y}$ is replaced with the identity matrix $\mathbf{I}$ in Eq 4. Then, the expression for the slope takes a simple form and provides the intuition for our novel parametric algorithm in section 2.6. From Eq 3, the sample covariance [22] is expressed as

$$\begin{aligned} \mathrm{Cov}(\mathbf{x}, \mathbf{y}) &= b_{xy}\mathrm{Var}(\mathbf{x}) + \mathrm{Cov}(\mathbf{x}, \mathbf{e_y} - b_{xy}\mathbf{e_x}), \\ &= b_{xy}(\mathrm{Var}(\mathbf{x}) - \mathrm{Var}(\mathbf{e_x})), \end{aligned}$$

where $\mathrm{Cov}(\mathbf{e_x}, \mathbf{e_y}) = 0$, and $\mathrm{Cov}(\mathbf{x}, \mathbf{e_x}) = \mathrm{Var}(\mathbf{e_x})$. Then, we obtain the direct estimator for the slope

$$\begin{aligned} b_{xy,\kappa_\mathbf{x}} &= \frac{\mathrm{Cov}(\mathbf{x}, \mathbf{y})}{\mathrm{Var}(\mathbf{x}) - \mathrm{Var}(\mathbf{e_x})}, \\ &= \frac{b_{xy,\mathrm{OLR}}}{\kappa_\mathbf{x}}, \end{aligned} \tag{9}$$

where

$$b_{xy,\mathrm{OLR}} \equiv \frac{\mathrm{Cov}(\mathbf{x}, \mathbf{y})}{\mathrm{Var}(\mathbf{x})} \tag{10}$$

and $b_{xy,\kappa_\mathbf{x}} \to b_{xy,\mathrm{OLR}}$ as $\kappa_\mathbf{x} \to 1$. Even though $b_{xy,\kappa_\mathbf{x}}$ is referred to as the 'regression coefficient corrected for attenuation' [17], we note that $b_{xy,\kappa_\mathbf{x}}$ is also biased. Reversing the roles of $\mathbf{x}$ and $\mathbf{y}$ yields the inverse estimator for $\beta_{xy}$:

$$\begin{aligned} b_{yx,\kappa_\mathbf{y}}^{-1} &= \frac{\mathrm{Var}(\mathbf{y}) - \mathrm{Var}(\mathbf{e_y})}{\mathrm{Cov}(\mathbf{y}, \mathbf{x})}, \\ &= \frac{\kappa_\mathbf{y}}{b_{yx,\mathrm{OLR}}}, \end{aligned} \tag{11}$$

where $\kappa_\mathbf{y} = 1 - \mathrm{Var}(\mathbf{e_y})/\mathrm{Var}(\mathbf{y})$. The relations in Eqs 9 and 11 indicate that the OLR estimates serve as bounding values because for any least squares estimate $b_{\mathrm{MER}}$, it can be shown [20] that

$$|b_{xy,\mathrm{OLR}}| \le |b_{\mathrm{MER}}| \le |b_{yx,\mathrm{OLR}}^{-1}|. \tag{12}$$

In section 2.6, we provide a statistical explanation (see Eq 26) and a graphical illustration (Fig 2) of the bounded range of $|b_{\mathrm{MER}}|$. To compensate for the bias in $b_{xy,\kappa_\mathbf{x}}$ and $b_{yx,\kappa_\mathbf{y}}^{-1}$, various ad-hoc combinations have been proposed, including the arithmetic mean [16, 35]:

$$b_{xy,\mathrm{A}} = (b_{xy,\kappa_\mathbf{x}} + b_{yx,\kappa_\mathbf{y}}^{-1})/2.$$

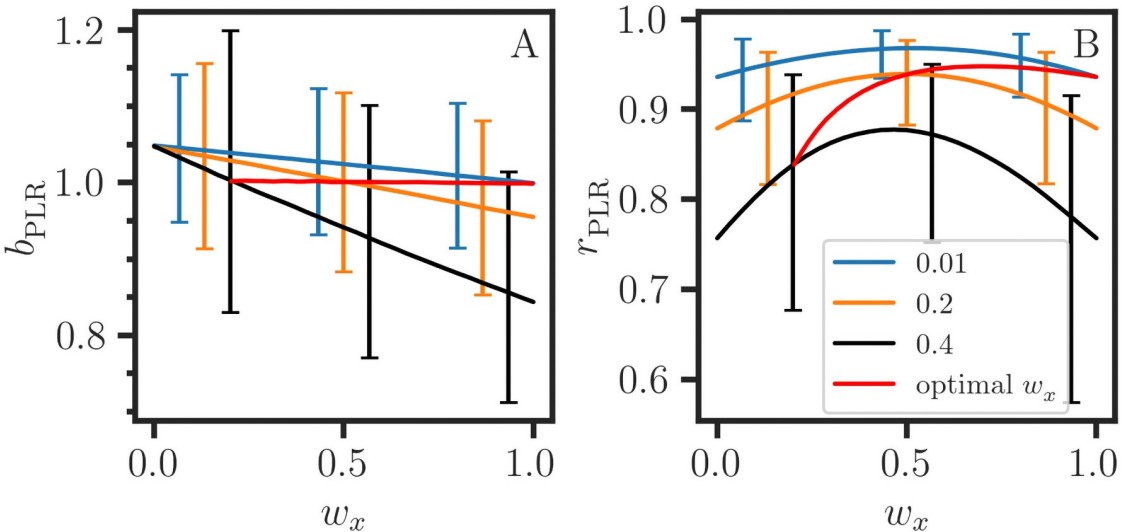

**Fig 2. Bounded ranges of the regression slopes for the errors in both x and y.** Monte Carlo simulations of parametric linear regression are conducted for 25 evenly spaced values $y_i^* = x_i^*$, and the error model is $s_{y_i} = 0.2 y_i^*$, $s_{x_i} = \alpha x_i^*$ for $\alpha = 0.01, 0.2, 0.4$. The estimates for the slope $b_{PLR}$ (A) and correlation $r_{PLR} = r_{\tau x} r_{\tau y}$ (B) are averages over 5000 datasets $(\mathbf{x}, \mathbf{y}) = (\mathbf{x}^* + \mathbf{e_x}, \mathbf{y}^* + \mathbf{e_y})$ with normally distributed errors $(\mathbf{e_x}, \mathbf{e_y})$ and a weighted average $\tau = w_x \mathbf{x} + (1 - w_x)\mathbf{y}$. The optimal $(b_{PLR}, r_{PLR})$ estimates corresponding to the minimum CVE in $\tau$ are also plotted (red). The $2\sigma$ error bars are estimates for the positive and negative deviations from the median value.

Unfortunately, the limiting values of $b_{xy,A}$ are not consistent with the OLR estimate for either $\kappa_{\mathbf{x}}$ or $\kappa_{\mathbf{y}} = 1$. Instead, we introduce the weighted average

$$b_{xy,W} = w_x \frac{b_{xy,WR}}{\kappa_{\mathbf{x}}} + w_y \frac{\kappa_{\mathbf{y}}}{b_{yx,WR}},$$

where the weights are determined from signal to noise ratios $(w_x, w_y) \propto (\bar{\mathbf{x}}/\text{Var}(\mathbf{e_x}), \bar{\mathbf{y}}/\text{Var}(\mathbf{e_y}))$ (see Eq 17). In Fig 1A and 1B, we observe that the $b_{xy,W}$ estimates compare well with the expected slope but with much larger confidence intervals compared to those of WOR.

## 2.5 Noise reduction for a weighted average of linearly related variable vectors

Consider the weighted average of the LRVVs $\{(\mathbf{x}, \mathbf{y}) \mid \mathbf{x}, \mathbf{y} \in \mathbb{R}^n\}$:

$$\tau = w\mathbf{x} + (1 - w)\mathbf{y}, \tag{13}$$

with $0 \leq w \leq 1$. Then, $\tau \in \mathbf{conv}(\mathbf{x}, \mathbf{y})$ and corresponds to points on the line segment connecting $\mathbf{x}$ and $\mathbf{y}$. The average of $\mathbf{x}$ and $\mathbf{y}$ is associated with the partial cancellation of error and an improved signal-to-noise ratio. Let $\text{Var}(\mathbf{e_x})$ and $\text{Var}(\mathbf{e_y})$ denote the expected variances of the errors in $\mathbf{x}$ and $\mathbf{y}$, respectively. Then, the propagation of independent error for the weighted sum of two variables [34] gives

$$\text{Var}(\mathbf{e_\tau}) = w^2 \text{Var}(\mathbf{e_x}) + (1 - w)^2 \text{Var}(\mathbf{e_y}). \tag{14}$$

By introducing the mean value $\bar{\tau} = w\bar{x} + (1-w)\bar{y}$ and the change in coordinates $\omega = w/(1-w)$, the coefficient of variation for error in $\tau$ is

$$\sqrt{\text{Var}(\mathbf{e}_\tau)}\bar{\tau} \quad = \frac{\sqrt{\dfrac{\text{Var}(\mathbf{e}_x)}{\text{Var}(\mathbf{e}_y)}\omega^2 + 1}}{\dfrac{\bar{x}}{\bar{y}}\omega + 1}\sqrt{\text{Var}(\mathbf{e}_y)}\bar{y}. \tag{15}$$

Then, setting the derivative $d(\sqrt{\text{Var}(\mathbf{e}_\tau)}/\bar{\tau})/d\omega = 0$ yields

$$\omega = \frac{\bar{x}}{\bar{y}}\frac{\text{Var}(\mathbf{e}_y)}{\text{Var}(\mathbf{e}_x)},$$

and we obtain the criterion

$$w = \left(1 + \frac{\bar{y}}{\bar{x}}\frac{\text{Var}(\mathbf{e}_x)}{\text{Var}(\mathbf{e}_y)}\right)^{-1}, \tag{16}$$

for the minimum CVE. In Fig 3, we use Monte Carlo simulations to demonstrate that the minimum $\sqrt{\text{Var}(\mathbf{e}_\tau)}/\bar{\tau}$ is consistent with Eq 16. By partitioning the weighted average $\boldsymbol{\tau} = \Sigma_j\, w_j\, \mathbf{y}_j$ into a sequence of pairwise averages for the data ($\mathbf{y}_j$), we obtain the general min(CVE) expression

$$w_j \quad = \frac{\bar{y}_j}{\text{Var}(\mathbf{e}_{y_j})}\left(\sum_j \frac{\bar{y}_j}{\text{Var}(\mathbf{e}_{y_j})}\right)^{-1}. \tag{17}$$

Thus, the noise reduction for the average is optimal when the $\bar{y}_j/\text{Var}(\mathbf{e}_{y_j})$ ratios are balanced for the LRVVs. To the best of our knowledge, this expression for optimizing the CVE of a weighted average has not been previously reported. We note that the $\bar{y}_j/\text{Var}(\mathbf{e}_{y_j})$ correspond to the signal-to-noise ratios $\bar{y}_j/\sqrt{\text{Var}(\mathbf{e}_{y_j})}$ [36]. However, for data that are not linearly related, averaging results in the uncontrolled mixing of noise and signal and a loss of information. This implies that the treatment of measurement error in more general problems, including multiple and polynomial regression, requires nonlinear optimization methods. That explains why this work is limited to the treatment of MER for LRVVs. In the next section, we discuss the use of $\tau$ as a parameter for measurement error regression.

## 2.6 The chain rule in linear regression

By applying the chain rule [34] to Eq 2, we obtain the factorization

$$\beta_{xy} \quad = \frac{dy_i^*}{dx_i^*}, \tag{18}$$

$$= \frac{dy_i^*}{d\tau_i^*}\frac{d\tau_i^*}{dx_i^*}, \tag{19}$$

$$\leftrightarrow \left(\frac{dy_i^*}{d\tau_i^*}, \frac{dx_i^*}{d\tau_i^*}\right), \tag{20}$$

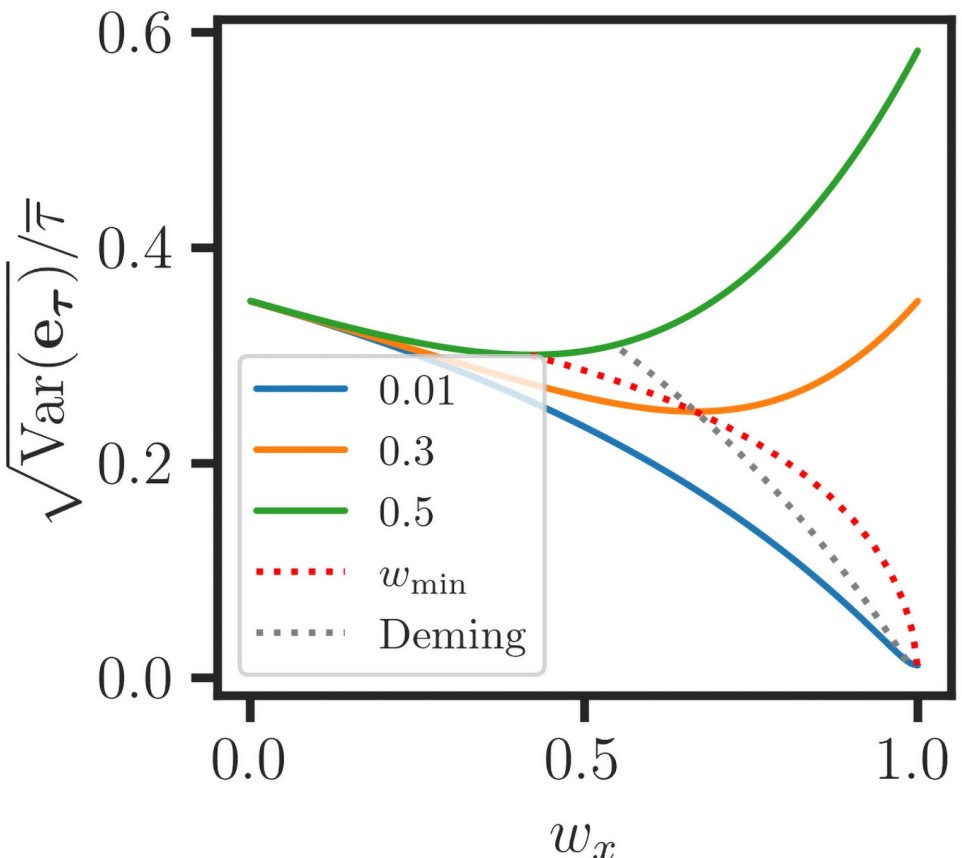

**Fig 3. Noise reduction for a weighted average of variable vectors.** Monte Carlo simulations of weighted averaging are conducted for the data $(\mathbf{x}, \mathbf{y})$ consisting of 25 evenly spaced values for $\mathbf{y}^* = 2\mathbf{x}^*$. The statistical parameters are estimated from 5000 datasets $(\mathbf{x}, \mathbf{y}) = (\mathbf{x}^* + \mathbf{e_x}, \mathbf{y}^* + \mathbf{e_y})$ with normally distributed errors $(\mathbf{e_x}, \mathbf{e_y})$, and the error model is: $\mathbf{s_{y^*}} = 0.3\mathbf{y}^*$, $\mathbf{s_{x^*}} = \alpha\,\mathbf{x}^*$ for $\alpha = 0.01, 0.3, 0.5$. The minimum coefficient of variation for error $\sqrt{\mathrm{Var}(\mathbf{e_\tau})}/\bar{\tau}$ for the weighted average $\boldsymbol{\tau} = w_x\mathbf{x} + (1-w_x)\mathbf{y}$ is consistent with Eq 17, as shown by the optimal $w_{\min}$ curve. Deming regression is associated with the weights $(w_x, w_y) \propto (1/\sqrt{\mathrm{Var}(\mathbf{e_x})}, 1/\sqrt{\mathrm{Var}(\mathbf{e_y})})$ and the corresponding CVE curve (grey) intersects the $w_{\min}$ curve.

$$\equiv \boldsymbol{\beta}_{xy} = (\beta_{\tau y}, \beta_{\tau x}). \tag{21}$$

The fractional form of Eq 19 is associated with scale invariance and corresponds to the homogeneous coordinates $\boldsymbol{\beta}_{xy}t$, with $t \in \mathbb{R}^1$. The mapping $\beta_{xy} \mapsto \boldsymbol{\beta}_{xy}$ is equivalent to invoking the parametric representation $(x(t), y(t))$ for the linear relation between the variables (Eq 8). Therefore, the chain rule permits a generalization in which the slope in linear regression is parameterized by $\tau^*$, where $\beta_{xy}$ corresponds to the point $(\beta_{xy}, 1)$ on the projective line $\mathbb{P}^1$ and a perspective function of the slope vector $\boldsymbol{\beta}_{xy}$ [4, 33]. Consequently, Eq 21 serves as the basis for our generalization of the linear regression algorithm, where the weighted average $\boldsymbol{\tau}$ serves to parameterize the relation between variable vectors. Analogously to Eq 9, the PLR slope vector is obtained as

$$\mathbf{b}_{\mathbf{xy},\mathrm{PLR}} \leftrightarrow \frac{1}{\kappa_\tau}\left(b_{\boldsymbol{\tau}\mathbf{y},\mathrm{WR}}, \, b_{\boldsymbol{\tau}\mathbf{x},\mathrm{WR}}\right), \tag{22}$$

where the components of $\mathbf{b}_{\mathbf{xy},\text{PLR}}$ are estimated individually using the WR algorithm (Eq 4) with $\boldsymbol{\tau}$ in the design matrix. Thus, $\mathbf{x}$ and $\mathbf{y}$ both serve as dependent variables. Without loss of generality, the factor $\kappa_{\boldsymbol{\tau}}^{-1}$ can be ignored because of the homogeneous coordinates equivalence. Therefore, we can regard $\boldsymbol{\tau}$ as corresponding to a set of fixed values. Then, the parametric linear regression estimator for $\beta_{xy}$ is

$$b_{xy,\text{PLR}} = \frac{b_{\boldsymbol{\tau}\mathbf{y},\text{WR}}}{b_{\boldsymbol{\tau}\mathbf{x},\text{WR}}}. \tag{23}$$

In the special case where $w_x = 1$, $\boldsymbol{\tau} = \mathbf{x}$ and $b_{xy,\text{PLR}} \to b_{xy,\text{WR}}$, $\mathbf{x}$ serves as both a dependent variable and an independent variable. Conversely, $b_{xy,\text{PLR}} \to b_{yx,\text{WR}}^{-1}$ for $w_y = 1$, and $\mathbf{y}$ serves as an independent variable. Therefore, the fact that there is a bounded range of values for the MER slope (Eq 12) comes from the dependence of $b_{xy,\text{PLR}}$ on $\boldsymbol{\tau}$, which ranges over $\mathbf{conv}(\mathbf{x}, \mathbf{y})$. The monotonic decrease of $b_{xy,\text{PLR}}$ with $w_x$ is shown in Fig 2A, with the lower and upper bounds given by $(b_{xy,\text{WR}}, 1)$ and $(1, b_{yx,\text{WR}})$, respectively; this is consistent with the OLR constraints in Eq 12. As we discuss in section 3.2, the extension of the PLR algorithm for estimating the MER parameters for a set of LRVVs $\boldsymbol{\mathcal{Y}} = (\mathbf{y}_j)$ is straightforward. Therefore, we make the *parametric linear regression* proposition.

**Proposition 1**. *For a set of linearly related variable vectors $\boldsymbol{\mathcal{Y}} = (\mathbf{y}_j)$, the weighted average $\boldsymbol{\tau}$ $= \Sigma_j w_j \mathbf{y}_j$ corresponding to the minimum coefficient of variation for error (Eq 17) provides the optimal measurement error regression estimate for the slope vector $\mathbf{b}_{\mathcal{Y},\text{PLR}} = (b_{\boldsymbol{\tau}\mathbf{y}_j,\text{WR}})$.*

In Fig 1A and 1B, we show that the performance of the $b_{xy,\text{PLR}}$ estimate is comparable to that of the WOR algorithm for a wide range of reliability ratios. The PLR algorithm provides predicted values for $\mathbf{y}_{(i)}$ (Fig 1E), but predicted values can also be estimated using the WP method (Eq 7). In the special case where the LRVVs are not subject to error and the points fall exactly on a straight line, we obtain $\kappa_{\boldsymbol{\tau}} = 1 \, \forall \, \boldsymbol{\tau} \in \mathbf{conv}(\boldsymbol{\mathcal{Y}})$, then any $\boldsymbol{\tau}$ can serve as the parametric variable vector in PLR.

Now, we consider the pedagogically important special case where $\mathbf{V}_{\mathbf{x}}$ and $\mathbf{V}_{\mathbf{y}}$ are both proportional to $\mathbf{I}$. We obtain

$$\mathbf{b}_{\mathbf{xy},\text{PLR}} \to \mathbf{b}_{\mathbf{xy},\text{POLR}} = \frac{1}{\text{Var}(\boldsymbol{\tau})}(\text{Cov}(\boldsymbol{\tau}, \mathbf{y}), \text{Cov}(\boldsymbol{\tau}, \mathbf{x})), \tag{24}$$

$$\leftrightarrow \mathbf{Cov}(\mathbf{y}, \mathbf{x} \mid \boldsymbol{\tau}) \equiv (\text{Cov}(\boldsymbol{\tau}, \mathbf{y}), \text{Cov}(\boldsymbol{\tau}, \mathbf{x})), \tag{25}$$

$$\leftrightarrow (b_{xy,\text{POLR}}, 1), \tag{26}$$

where $b_{xy,\text{POLR}} = \text{Cov}(\boldsymbol{\tau}, \mathbf{y})/\text{Cov}(\boldsymbol{\tau}, \mathbf{x})$ is a perspective function of $\mathbf{b}_{\mathbf{xy},\text{POLR}}$. In Eq 25, we introduce the parametric covariance vector notation $\mathbf{Cov}(\mathbf{y}, \mathbf{x} \mid \boldsymbol{\tau})$ to indicate the homogeneous coordinates equivalence with $\mathbf{b}_{\mathbf{xy},\text{POLR}}$. Thus, we note that covariance serves as a measure of linear dependence between variable vectors [21, 32] and is therefore subject to the chain rule. Consequently, we make the *parametric covariance* proposition.

**Proposition 2** *In the analysis of linear dependence for a set of LRVVs $\boldsymbol{\mathcal{Y}}$, the chain rule implies that the covariance is associated with a conditional vector form* $\mathbf{Cov}(\boldsymbol{\mathcal{Y}} \mid \boldsymbol{\tau}) = (\text{Cov}(\boldsymbol{\tau}, \mathbf{y}_j))$, *which is parameterized by the weighted average $\boldsymbol{\tau} \in \mathbf{conv}(\boldsymbol{\mathcal{Y}})$.*

Therefore, covariance is a vector quantity in the parametric representation for the linear relation between variable vectors. Then, there is a range for $\mathbf{Cov}(\mathbf{y}, \mathbf{x} \mid \boldsymbol{\tau})$ in correspondence with $\mathbf{b}_{\mathbf{xy},\text{POLR}}$ and the bounded range for $b_{\text{MER}}$ in Eq 12. For $(w_x, w_y)$ given by Eq 17 and $\boldsymbol{\tau} = w_x$

$\mathbf{x} + w_x \mathbf{y}$, we obtain the expression

$$\frac{\text{Cov}(\boldsymbol{\tau}, \mathbf{y})}{w_y \text{Var}(\mathbf{y})} = \frac{w_x \text{Cov}(\mathbf{x}, \mathbf{y})}{w_y \text{Var}(\mathbf{y})} + 1, \tag{27}$$

observe that the right side is invariant to scaling the data: $(\mathbf{x}, \mathbf{y}) \rightarrow (k_x\mathbf{x}, k_y\mathbf{y})$, and note that there is a similar factorization for $\text{Cov}(\boldsymbol{\tau}, \mathbf{x})$. Conversely, we conclude that the scaling invariance property for the quantities

$$\left\{ \frac{w_x \text{Cov}(\mathbf{x}, \mathbf{y})}{w_y \text{Var}(\mathbf{y})}, \frac{w_y \text{Cov}(\mathbf{y}, \mathbf{x})}{w_x \text{Var}(\mathbf{x})} \right\},$$

implies that $w_x/w_y \propto k_y/k_x$, which is a necessary condition but not sufficient for specifying the optimal weights for $\boldsymbol{\tau}$; in section 3.1, we discuss Deming weighting which also satisfies the scaling invariance property. Then, we obtain the homogeneous coordinates equivalence

$$(\text{Cov}(\boldsymbol{\tau}, \mathbf{y}), \text{Cov}(\boldsymbol{\tau}, \mathbf{x})) \leftrightarrow \left( \frac{w_y \text{Var}(\mathbf{y})}{w_x \text{Var}(\mathbf{x})} \frac{\text{Cov}(\boldsymbol{\tau}, \mathbf{y})}{w_y \text{Var}(\mathbf{y})}, \frac{\text{Cov}(\boldsymbol{\tau}, \mathbf{x})}{w_x \text{Var}(\mathbf{x})} \right),$$

and observe the corresponding scaling relation

$$\frac{w_y \text{Var}(\mathbf{y})}{w_x \text{Var}(\mathbf{x})} \rightarrow \frac{k_y}{k_x} \frac{w_y \text{Var}(\mathbf{y})}{w_x \text{Var}(\mathbf{x})}.$$

Then, $\mathbf{b}_{\mathbf{xy},\text{POLR}}$ varies proportionally with the scaling $(k_x\mathbf{x}, k_y\mathbf{y})$; the $\mathbf{b}_{\mathbf{xy},\text{POLR}}$ estimate is consistent with respect to the scaling of the data. These covariance factorizations provide another demonstration of the importance of WLS optimization for partitioning residual effects in measurement error regression. The generalization of these expressions for a $(\mathbf{y}_j)$ dataset is straightforward. However, we note that $\text{Cov}(\boldsymbol{\tau}, \mathbf{y}_j)$ assumes that $\mathbf{V}_{\mathbf{y}_j} = \mathbf{I}$ and does not account for the heteroscedastic error in $\mathbf{y}_j$. Thus, in practice, the PLR estimate (Eq 22) is preferred. Finally, we note that the two special cases

$$\mathbf{b}_{\mathbf{xy},\text{POLR}} = (b_{xy,\text{OLR}}, 1), \tag{28}$$

$$\leftrightarrow (\text{Cov}(\mathbf{x}, \mathbf{y}), \text{Var}(\mathbf{x})), \tag{29}$$

for $\boldsymbol{\tau} = \mathbf{x}$ and

$$\mathbf{b}_{\mathbf{xy},\text{POLR}} = (1, b_{yx,\text{OLR}}), \tag{30}$$

$$\leftrightarrow (\text{Var}(\mathbf{y}), \text{Cov}(\mathbf{y}, \mathbf{x})), \tag{31}$$

for $\boldsymbol{\tau} = \mathbf{y}$, correspond to the bounding values in Eq 12. Therefore, $\text{Cov}(\mathbf{x}, \mathbf{y})$ and $\text{Cov}(\mathbf{y}, \mathbf{x})$ are numerically equal but serve as components for different parametric covariance vectors, and correspond to different estimates for MER slope, $b_{xy,\text{OLR}}$ and $b_{yx,\text{OLR}}$, respectively.

Now, we consider the multivariate generalization of the Pearson correlation coefficient $r$ (Eq 1). Using the well-known correspondence between covariance and Pearson correlation [32],

$$r_{\boldsymbol{\tau}\mathbf{y}_j} = \frac{\text{Cov}(\boldsymbol{\tau}, \mathbf{y}_j)}{\sqrt{\text{Var}(\boldsymbol{\tau})\text{Var}(\mathbf{y}_j)}} = \hat{\boldsymbol{\tau}}_c \cdot \hat{\mathbf{y}}_{j,c},$$

we transform Eq 25 and define a parametric correlation vector

$$\mathbf{r}(\mathbf{x}, \mathbf{y} \mid \boldsymbol{\tau}) = (r_{\boldsymbol{\tau}\mathbf{x}}, r_{\boldsymbol{\tau}\mathbf{y}}). \tag{32}$$

Then, we observe that the Pearson correlation coefficient $r_{\boldsymbol{\tau}}(\mathbf{x}, \mathbf{y})$ can be written in Cartesian tensor form as the double contraction of dyads

$$r_{\boldsymbol{\tau}} = \hat{\boldsymbol{\tau}}_c \hat{\boldsymbol{\tau}}_c : \hat{\mathbf{x}}_c \hat{\mathbf{y}}_c, \tag{33}$$

$$= r_{\boldsymbol{\tau}\mathbf{x}} r_{\boldsymbol{\tau}\mathbf{y}}. \tag{34}$$

The dyad $\hat{\mathbf{x}}_c \hat{\mathbf{y}}_c$ is a bilinear measure of the joint variation in $\mathbf{x}$ and $\mathbf{y}$. In the special cases where $\hat{\boldsymbol{\tau}}_c = \hat{\mathbf{x}}_c$ and $\hat{\mathbf{y}}_c$, $r_{\boldsymbol{\tau}}$ reduces to the standard Pearson correlation coefficient $r$ (Eq 1). Geometrically, $r_{\boldsymbol{\tau}}$ serves as a measure of the degree to which $\mathbf{x}_c$ and $\mathbf{y}_c$ are mutually aligned in the $\boldsymbol{\tau}_c$ direction. Then, the generalization of Eq 33 as the $m$-fold contraction of polyads for the variable vectors $(\hat{\mathbf{y}}_{j,c}) \in \mathbb{R}^{n \times m}$ serves as the basis for our definition of the parametric correlation coefficient:

$$r_{\text{PLR}} = \prod_{j=1}^{m} r_{\boldsymbol{\tau}\mathbf{y}_j}, \tag{35}$$

with $m \geq 2$. The $\hat{\boldsymbol{\tau}}_c$-polyads $\hat{\boldsymbol{\tau}}_c \hat{\boldsymbol{\tau}}_c$, $\hat{\boldsymbol{\tau}}_c \hat{\boldsymbol{\tau}}_c \hat{\boldsymbol{\tau}}_c$, ... are referred to as $2^{\text{nd}}$, $3^{\text{rd}}$, ... order Pearson correlation tensors, respectively. Therefore, we associate a set of $m$ LRVVs with a set of two-way, three-way, ..., $m$-way weighted averages, PLRs, covariance vectors and PLR correlation coefficients. However, a detailed discussion of Cartesian tensors [37, 38] is beyond the scope of this paper. In Fig 2B, the concave down variation of $r_{\text{PLR}} = r_{\boldsymbol{\tau}\mathbf{x}} r_{\boldsymbol{\tau}\mathbf{y}}$ results from the product of the monotonically increasing $r_{\boldsymbol{\tau}\mathbf{x}}$ function and decreasing $r_{\boldsymbol{\tau}\mathbf{y}}$ function, with the endpoints equal to the Pearson correlation coefficient $r$. Thus, $r_{\text{PLR}}$ varies with the expected error $(\mathbf{s}_{\mathbf{x}}^2, \mathbf{s}_{\mathbf{y}}^2)$, but Pearson's $r$ serves as the lower bound. Then, $r$ is attenuated compared to $r_{\text{PLR}}$ when the data $(\mathbf{x}, \mathbf{y})$ are subject to measurement errors [13], as shown in Fig 1C. When the data $(\mathbf{x}, \mathbf{y})$ are highly correlated and Pearson's $r$ approaches 1, the $r_{\text{PLR}}$ curve becomes a horizontal line at $r_{\text{PLR}} = 1$ (or nearly so). We conclude that the parametric covariance (Eq 25), correlation (Eq 32) and PLR slope (Proposition 1) correspond to different perspective functions and coordinate systems for representing the linear relation between variable vectors, i.e., points dispersed about the MER line. Then, a condition for the interpretation of covariance and correlation is that the deviation of the points from the MER line must be consistent with the expected error $(\mathbf{s}_{\mathbf{y}_{(i)}}^2)$. In section 3.1, we discuss the ambiguity that arises when the MER line is perturbed by outliers.

Analytical methods for the propagation of errors and the estimation of distributions for ratios [39, 40], proportions [41, 42], and correlation coefficients [43] are complicated by fractional transformations, bounded ranges, and discreteness. The propagation of errors for convoluted quantities such as $b_{xy,\text{PLR}}$ and $r_{\text{PLR}}$ is prohibitively complicated. Alternatively, Monte Carlo methods are practical approaches for estimating the distributions and confidence intervals of fractional statistical parameters, and they allow for the detailed simulation of stochastic effects in the data acquisition process as described previously [4]. The error bars shown in various figures in this paper are estimated using MC methods. Fig 4 shows an example of a two-dimensional MC histogram for the joint $(r_{\text{PLR}}, b_{xy,\text{PLR}})$ distribution. We note that the distributions for MER parameters are often skewed, so confidence intervals are estimated separately for $+/-$ deviations from the median.

## 3 Data analysis and results

Using publicly available data, we provide practical examples that illustrate the ambiguity that arises when the variable vectors are not linearly related and the application of multidimensional parametric linear regression for analyzing the dispersion of RNA-Seq read count data.

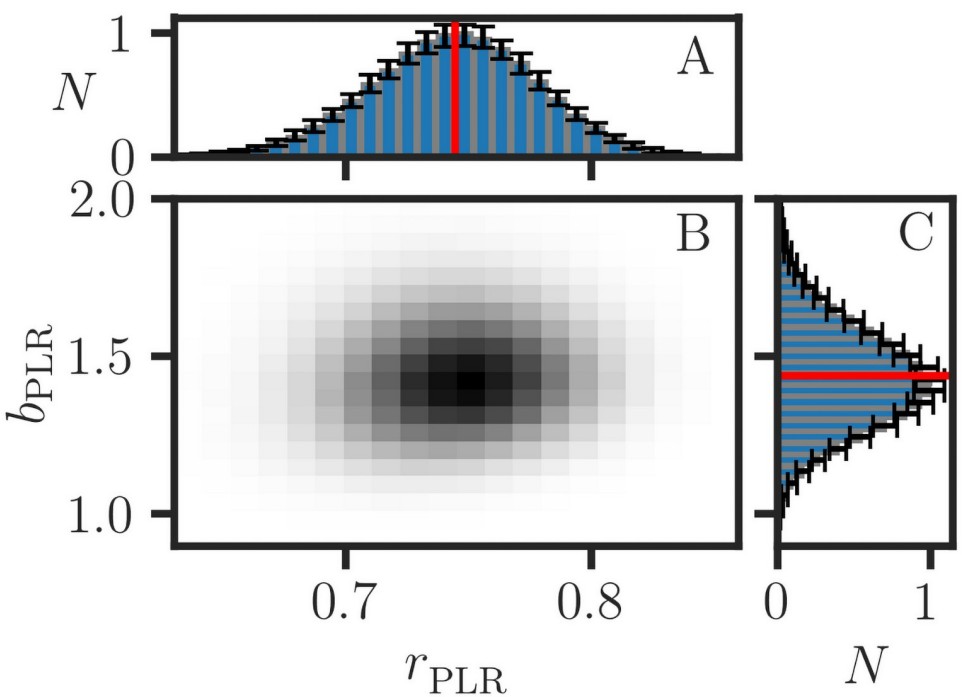

**Fig 4. Distribution of the statistical parameters in parametric linear regression.** A two-dimensional histogram (B) is calculated from 500000 Monte Carlo datasets for the data in Fig 1D. The distributions of $r_{\mathrm{PLR}}$ (A) and $b_{\mathrm{PLR}}$ (C) are averages obtained from 100 MC histograms, with 5000 MC datasets ($\mathbf{x}$, $\mathbf{y}$) per histogram. The $2\sigma$ error bars are estimates of the positive and negative deviations from the median and indicate the degree of convergence of the MC estimate. The $(r_{\mathrm{PLR}}, b_{\mathrm{PLR}}) = (0.74, 1.44)$ estimates are indicated in red.

### 3.1 Example 1: Measurement error regression with outliers

The data for this example come from Table 11.3.1 in [21] and originate from consulting work (R. Berger, personal communication). However, error estimates are not available for these data. For our demonstration, we provided a rudimentary error model with $(\mathbf{s}_{\mathbf{x}}^2, \mathbf{s}_{\mathbf{y}}^2) = (0.04 + 0.005\mathbf{x}^2, 0.1 + 0.005\mathbf{y}^2)$, as shown by the $2\sigma$ xy-error bars in Fig 1D. The ordinary linear regression estimate for the slope is attenuated as expected (Eq 9). However, the orthogonal regression (OR) estimate reported in [21] is also biased because of the equal weight assumption for the residual effects ($\mathbf{e_x}$, $\mathbf{e_y}$). Using the defined notations for sums of squares,

$$S_{xx} = \sum_i (x_i - \bar{x}_i)^2, \quad S_{yy} = \sum_i (y_i - \bar{y}_i)^2, \quad S_{xy} = \sum_i (x_i - \bar{x}_i)(y_i - \bar{y}_i),$$

and making the substitution $(\mathbf{x}, \mathbf{y}) \mapsto (w_x\mathbf{x}, w_y\mathbf{y})$, we obtain a WLS OR estimate:

$$b_{xy,\mathrm{ORw}} = \frac{w_x}{w_y}\left(-B + \sqrt{B^2 + 1}\right), \tag{36}$$

where the $w_x/w_y$ factor is needed for the transformation back to the original coordinates for ($\mathbf{x}$, $\mathbf{y}$), and the factor for the weighted residuals in ($\mathbf{x}$, $\mathbf{y}$) is

$$B = \frac{1}{2S_{xy}}\left(\frac{w_x}{w_y}S_{xx} - \frac{w_y}{w_x}S_{yy}\right). \tag{37}$$

The weights are determined from the signal to noise ratios $(w_x, w_y) \propto$ $(\bar{\mathbf{x}}/\mathrm{Var}(\mathbf{e_x}), \bar{\mathbf{y}}/\mathrm{Var}(\mathbf{e_y}))$ with the constraints $w_x + w_y = 1$ and $w_x, w_y \geq 0$ (Eq 17). Then, we note that $B$ is invariant to the scaling of the data, $(\mathbf{x}, \mathbf{y}) \rightarrow (k_x \mathbf{x}, k_y \mathbf{y})$. Therefore, the $b_{xy,\mathrm{ORw}}$ estimate is consistent with respect to the scaling of the data. The $b_{xy,\mathrm{OR}}$ estimate is recovered when $w_x = w_y$ [20], and because of this constraint, $b_{xy,\mathrm{OR}}$ is not a consistent estimate since it does not vary proportionally with $(k_x \mathbf{x}, k_y \mathbf{y})$ scaling. There is better agreement between $b_{xy,\mathrm{WOR}}$ and $b_{xy,\mathrm{ORw}}$ (Fig 1D), but the latter does not account for the heteroscedastic errors $(\mathbf{s_x^2}, \mathbf{s_y^2})$. The $b_{xy,\mathrm{PLR}}$ estimate is obtained from the separate regressions $(b_{\mathbf{rx},\mathrm{WR}}, b_{\mathbf{ry},\mathrm{WR}})$, as shown in Fig 1E. The differences between the $b_{xy,\mathrm{WOR}}$ and $b_{xy,\mathrm{PLR}}$ estimates are small compared to the experimental uncertainty. However, we observe that there are many positive and negative outliers in the data (Fig 1D), and only a subset of the points are close to any one of the MER lines. This implies that the data $(\mathbf{x}, \mathbf{y})$ are not linearly related, and the weighted average $\boldsymbol{\tau} \in \mathbf{conv}(\mathbf{x}, \mathbf{y})$ is then subject to confounding effects. Then, WLS optimization produces misleading results, and the MER line does not possess a functional or operational interpretation. Criteria can be applied to select an LRVV subset for measurement error regression, but this works best when the outlier subset is small. Otherwise, linear regression is not recommended for the analysis of data that are not linearly related. In [4], we discuss the use of regression tree association graphs in the analysis of weakly correlated data. Finally, we note that the Deming orthogonal regression estimate [22] is obtained by substituting the alternative weighting $(w_x, w_y) \propto$ $(1/\sqrt{\mathrm{Var}(\mathbf{e_x})}, 1/\sqrt{\mathrm{Var}(\mathbf{e_y})})$ into Eq 36. Our simulations confirm that Deming weighting has the scale invariance property but it is not optimal for MER (Fig 3), and the corresponding parameter estimates are biased for both PLR and ORw (not shown).

## 3.2 Example 2: Conical dispersion for replicated RNA-Seq data

In this section, our objective is to demonstrate that multidimensional linear MER has practical applications. We expect that there are many applications for PLR in the 'big data' world but the application for RNA-Seq data analysis is convenient because of our previous collaborations in gene expression research. In particular, we discuss the use of the PLR method in the analysis of read counting errors in a replicated RNA-Seq dataset. The estimation of the experimental errors and confidence intervals for effect size measures are important problems in RNA-Seq analysis because experiments are usually performed with very few replicates [6]. The replicated data correspond to the Cartesian product of LRVVs $\boldsymbol{\mathcal{Y}} = (\mathbf{y}_j)$ with $\mathbf{y}_j \in \mathbb{R}^n$, $\dim(\boldsymbol{\mathcal{Y}}) = n \times m$; $y_{ij}$ is the read count for the $i^{\text{th}}$ RNA tag of the $\mathbf{y}_j$ replicate, $n$ is the number of RNA tags, $m$ is the number of replicates, and $m \ll n$ because RNA-Seq assays are highly multiplexed and $n$ is large. The observation vectors for the RNA tags are denoted $\mathbf{g}_{(i)}$. Our error analysis algorithm is iterative and requires an initial guess for the expected error in the data $(\mathbf{s}_{\mathbf{y}_j}^2)_{\mathrm{init}}$; the read counting errors are assumed to be independent or approximately so: $\{\mathrm{Cov}(e_{y_{ij}}, e_{y_{kl}}) = 0 \mid i \neq k \vee j \neq l\}$. In our implementation, the initial error estimate is of the form $\mathbf{s}_{\mathbf{y}_j}^2 = q_0 + q_2 \mathbf{y}_j^2$. Let $\min(\mathbf{g}_{(i)})$ correspond to a set of RNA tags with the lowest read counts, excluding those with $\bar{\mathbf{g}}_{(i)} = 0$; i.e., $\min(\mathbf{g}_{(i)})$ is a set of points close to the origin of the regression graph. Then, $q_0$ is initially estimated from the variance for $\min(\mathbf{g}_{(i)})$; typically $1 \leq q_0 \leq 50$. An initial value for $q_2$ is obtained by visual inspection of the data (Fig 5); typically $0.002 \leq q_2 \leq 0.05$ (Fig 6 legend). Then, the optimal $\boldsymbol{\tau} \in \mathbf{conv}(\boldsymbol{\mathcal{Y}})$ for the minimum CVE is estimated, and the PLR method provides parameter estimates for the MER line:

$$\ell = \mathbf{a}_{\mathrm{min}} + \mathbf{b}_{\mathrm{PLR}} t, \tag{38}$$

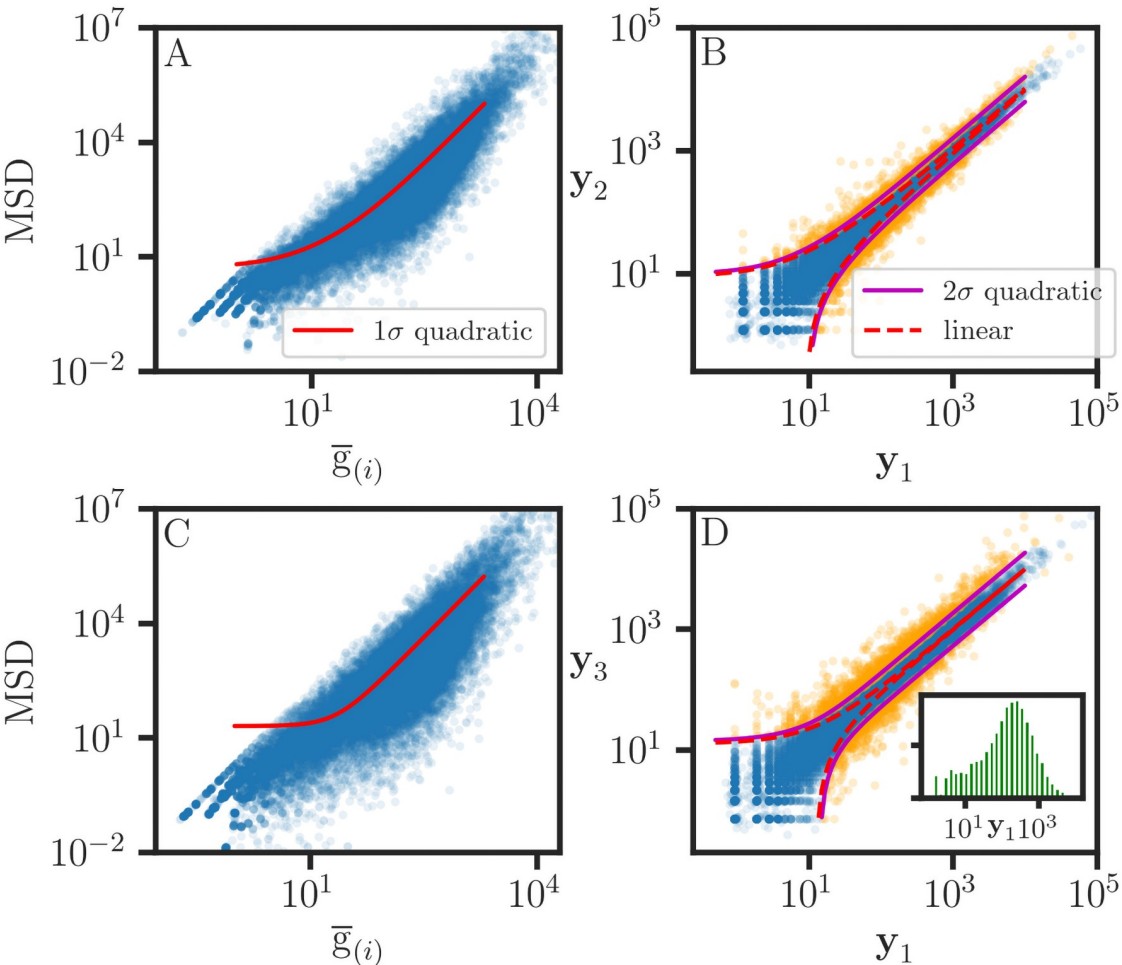

**Fig 5. Conical quadratic error parameters for replicated RNA-Seq data.** Replicate *Arabidopsis thaliana* RNA-Seq data from [46] for the ($\mu g$, 4 reps) sample in (A, B) and ($1g$, 3 reps) sample in (C,D) are scaled so that the parametric linear regression line corresponds to the unit slope line ($\mathbf{b}_{\mathrm{PLR}} = \mathbf{1}$) and passes through the origin. Then, the conical dispersion of points about the **1** line in the MER graph serves as a measure of the replication error, as shown in the $\mathbf{y}_1$ projections (B,D). An iterative WLS regression of the mean squared deviation versus the mean value ($\bar{\mathbf{g}}_{(i)}$) produces parameter estimates for the quadratic error model (Eq 41), as shown in (A, C). The corresponding $2\sigma$ threshold curves are shown in (B, D). The linear component of the error ($q_0 + q_1\bar{\mathbf{g}}_{(i)}$) is shown as 'convergent' dashed curves. The larger overdispersion $q_2$ in (C, D) compared to that in (A, B) indicates that there is a significant variation in data quality between the two experiments. The histogram (D inset) shows that the read counts range over four decades.

with $\mathbf{a}_{\mathrm{min}} = \overline{\min(\mathbf{g}_{(i)})}$, $t \in \mathbb{R}^1$, $\{\mathbf{a}_{\mathrm{min}}, \mathbf{b}_{\mathrm{PLR}} \in \mathbb{R}^m\}$. Then, the data are transformed so that $\boldsymbol{\ell}$ passes through the origin with a unit slope (Fig 5B & 5D); i.e., $\mathbf{b}_{\mathrm{PLR}} = \mathbf{1}$, and $\mathbf{a}_{\mathrm{min}} \cong \mathbf{0}$. This adjustment for $\boldsymbol{\ell}$ is important because afterwards, the $\mathbf{g}_{(i)}$ are conically dispersed about the **1** line. Then, the $\mathbf{g}_{(i)}$ are partitioned into orthogonal components for deviation $\boldsymbol{\delta}_{(i)}$ and mean $\bar{\mathbf{g}}_{(i)} = \mathbf{g}_{(i)} \cdot \mathbf{1}/\mathrm{dim}(\mathbf{g}_{(i)})$:

$$\mathbf{g}_{(i)} = \boldsymbol{\delta}_{(i)} + \bar{\mathbf{g}}_{(i)}\mathbf{1}. \tag{39}$$

Statistically, only the magnitude of $\boldsymbol{\delta}_{(i)}$ is important because the orientation is random. Next, we introduce the mean squared deviation (MSD) $\overline{\delta^2_{(i)}} = \boldsymbol{\delta}_{(i)} \cdot \boldsymbol{\delta}_{(i)}/\mathrm{dim}(\boldsymbol{\delta}_{(i)})$, and we obtain the two parameter representation for the dispersion ($\bar{\mathbf{g}}_{(i)}, \overline{\delta^2_{(i)}}$), as shown in Fig 5A and

5C. We adopt a heuristic quadratic model for the MSD [14, 44]

$$\overline{\delta_{(i)}^2} = q_0 + q_1 \bar{g}_{(i)} + q_2 \bar{g}_{(i)}^2 + e_{(i)}. \tag{40}$$

Then, a WLS quadratic regression of the MSD produces sample estimates for the $q_i$ (Fig 5A and 5C), which then serve as the error model for the input data:

$$s_{y_{ij}}^2 = q_0 + q_1 y_{ij} + q_2 y_{ij}^2, \tag{41}$$

as shown by the $2\sigma$ threshold curves in Fig 5B and 5D. The $q_0$ and $q_1$ terms correspond to the background noise and Poisson-like components of the dispersion, respectively [45]. The over-dispersive component $q_2$ suggests that there are uncontrolled experimental effects that cannot be averaged in a random manner. We note that for a well-designed data acquisition process, the $q_2$ component should be minimized. The initial guess for $(s_{y_j}^2)_{\mathrm{init}}$ is updated, and the algorithm iterates several times to reach convergence for both $\ell$ and $\{q_i\}$.

The data for this demonstration come from a study of the response of *Arabidopsis thaliana* to fractional gravity [47] and can be retrieved from the publicly accessible NASA GeneLab Omics database [46]. The dataset contains read count data for six samples with varying gravities ($g$) $\{\mu, 0.09, 0.18, 0.36, 0.57, 1\}$. For each sample, the replicates ($\mathbf{y}_j$) with $m = 3 \vee 4$, correspond to a set of LRVVs with approximately identical noise properties: $\{\mathbf{s}_{\mathbf{y}_j}^2 \cong \mathbf{s}_{\mathbf{y}_k}^2 \mid j \neq k\}$, with the possible exception of a small subset of irreproducible outliers which must be flagged. The read counts are distributed over more than three decades (Fig 5D inset), and the point estimates are skewed due to the heavy tail effects at large values [6]. Consequently, an upper limit for the mean value ($\bar{\mathbf{g}}_{(i)} \leq 2000$) of the observation vectors is imposed, and 4% of the data are removed from the MER analysis; however, all of the data are included in the graphs (Figs 5 and 6). In Fig 5, we observe that the dispersion varies between the $\mu g$ (A, B) and $1g$ (C, D) samples. As discussed above, the data are transformed such that the PLR slope is 1 and the curvature is negligible near the origin in log-log plots (B, D). We note that log-log scales are convenient for visualizing dispersion, but the data are not log-transformed in actual calculations; a log or ratio transformation results in nonlinear confounding effects for weighted averages. For duplicate read counts $\mathbf{g}_{(i)} = (x, y)$, we define the scaled difference

$$d = \frac{x - y}{\sqrt{s_x^2 + s_y^2}}, \tag{42}$$

with the expected variance for the error in $x - y$ given by the sum of the individual variances: $s_{x-y}^2 = s_x^2 + s_y^2$ [21]. Then, we apply the condition $s_y^2 = q_0 + q_1 y + q_2 y^2$ to obtain the quadratic equation

$$\left(q_2 - \frac{1}{d^2}\right) y^2 + \left(q_1 + \frac{2x}{d^2}\right) y + q_0 + s_x^2 - \frac{x^2}{d^2} = 0. \tag{43}$$

The $|d| = 2$ threshold curves in Fig 5B and 5D are calculated from the roots of this expression. The $q_2$ component dominates at high read counts, as indicated by the convergence of the dashed curves for the linear components ($q_0$, $q_1$). Then, $q_2$ serves as an indicator of the data quality and reproducibility, and we observe significant variations between the samples (Fig 6

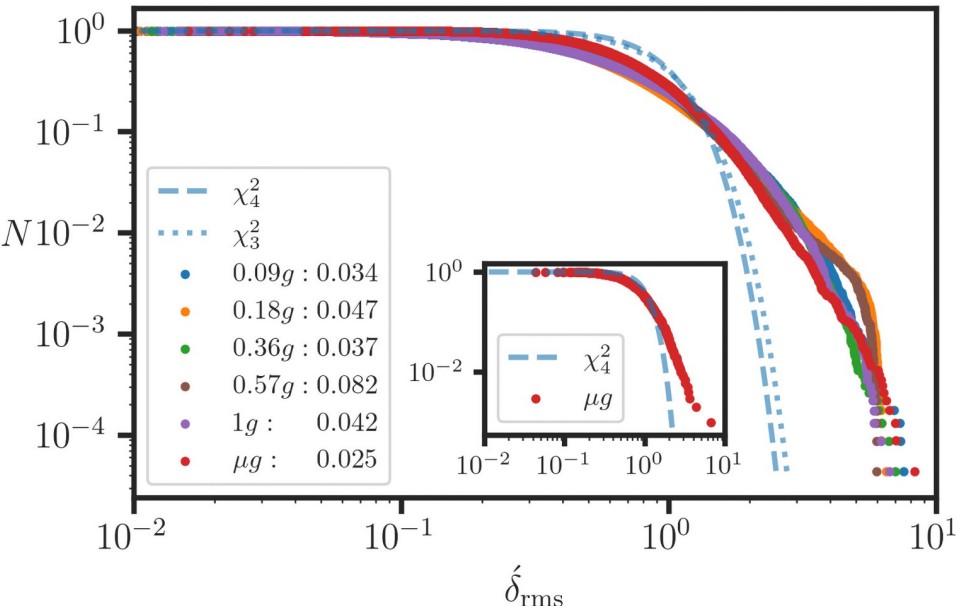

**Fig 6. Overdispersion of the residual effects in replicated RNA-Seq data.** Empirical survival functions for the root mean square scaled deviation ($\acute{\delta}_{\mathrm{rms}}$) from the conical dispersion analysis (see Fig 5) of the RNA-Seq data are plotted for six *A. thaliana* samples under fractional gravity [46]. The $\chi_3^2$ and $\chi_4^2$ curves from the Monte Carlo simulation for the standard normal distribution are also shown. The slower decay in the observed $\acute{\delta}_{\mathrm{rms}}$ indicates that RNA-Seq data are subject to heterogeneous dispersive effects. The inset shows the overdispersion for a 1000 point subset around the median $\bar{\mathbf{g}}_{(i)}$ for the $\mu g$ sample. Thus, overdispersive effects are observed across the entire range of read counts. In the legend, each sample is labeled with its $q_2$ coefficient for the quadratic error model. The variation in $q_2$ indicates that there are significant data quality variations between samples.

legend). We define the root mean square (rms) scaled deviation as

$$\acute{\delta}_{\mathrm{rms},(i)} = \sqrt{\frac{\overline{\delta_{(i)}^2}}{s_{\bar{\mathbf{g}}_{(i)}}^2}},$$

where $s_{\bar{\mathbf{g}}_{(i)}}^2 = q_0 + q_1 \bar{\mathbf{g}}_{(i)} + q_2 \bar{\mathbf{g}}_{(i)}^2$. A comparison of the empirical survival distributions for $\acute{\delta}_{\mathrm{rms},(i)}$ shows that there are heavy tails at large deviations for the six samples (Fig 6). We use Monte Carlo methods to simulate the read count data for normal distributions $\mathcal{N}(\bar{\mathbf{g}}_{(i)}, s_{\bar{\mathbf{g}}_{(i)}}, \dim(\mathbf{g}_{(i)}))$ and confirm that the simulated distributions for $\acute{\delta}_{\mathrm{rms},(i)}$ are identical to chi-squared distributions (Fig 6). The much slower decay in the experimental $\acute{\delta}_{\mathrm{rms},(i)}$ is an indication of uncontrolled processes in the RNA-Seq assay that result in overdispersion in the $\mathbf{g}_{(i)}$. The inset graph in Fig 6 shows the overdispersion for a small $\{\mathbf{g}_{(i)}\}$ subset around the median $\bar{\mathbf{g}}_{(i)}$ for the $\mu g$ sample. Thus, the overdispersive effects in the $\acute{\delta}_{\mathrm{rms},(i)}$ are observed for intervals over the entire range of read counts. Statistical methods that assume normal distributions and do not account for this overdispersion will overestimate the effect size. In biological studies, the variable vectors for RNA-Seq are not linearly related. Then, our parametric LRVV framework serves an elementary role in searching for multiway dependency in the variation of gene expression. The detailed discussion of weighted least squares optimization problems, including normalization and the estimation of effect size, in gene expression data analysis is a topic for a separate paper.

## 4 Software

Numerical computations were performed using the python (v3.8.4) language with the NumPy (v1.19.0) [48] and SciPy (v1.5.0) [49] packages. Figures were prepared using the Matplotlib (v3.4.1) package [50]. The manuscript was prepared using the MiKTeX (v2.9) implementation of TeX/LaTeX [51].

## 5 Discussion

In this work, we discuss the fact that statistical measures of linear dependence, including covariance, correlation coefficient and regression slope, are subject to the chain rule. We develop a novel multidimensional linear regression algorithm where the relation between variable vectors is parameterized by a weighted average, and the weights are determined from an error model $\mathcal{E}$ for the input data. Then, the implementation of PLR involves the use of weighted least squares normal equations to estimate the parameters of the best fit line for a set of LRVVs. Standard statistical concepts, including covariance and the Moore-Penrose pseudoinverse, provide the necessary intuition for the formulation of our PLR method. This contrasts with the ODRPACK algorithm which requires iterative numeric optimization because the WOR least-squares equation cannot be solved analytically. The PLR and ODRPACK methods yield very similar results but the latter is formulated only for bivariate data. We find that in the parametric representation of a linear relation, covariance and correlation are associated with vector quantities. We also identify scale invariant quantities for partitioned residual effects between the LRVVs for both the PLR and weighted orthogonal regression methods. Therefore, when $\mathcal{E}$ is undefined, MER is not a well-posed problem [52]; i.e., there is not a unique solution. This supports previous suggestions that the specification of $\mathcal{E}$ is a requirement for solving the MER problem [27]. $\mathcal{E}$ is also essential for determining the distributions and corresponding confidence intervals of PLR parameter estimates and effect size. Constructing a realistic error model for a data acquisition process can be daunting, especially in omics studies, because it is necessary to account for all relevant sources of experimental error and system variability. However, an approximate trial and error approach can be informative, and Monte Carlo methods are convenient because they allow for the detailed simulation of stochastic effects in the data acquisition process. Finally, the averaging of variable vectors that are not linearly related results in the loss of information because of the destructive interference associated with the mixing of signal and noise. Consequently, accounting for measurement error in polynomial and multiple regression requires nonlinear optimization methods, which is a topic for a separate paper. We conclude that the parametric representation for the relation between variable vectors provides a more general framework for linear regression compared to the standard Cartesian representation.

## Acknowledgments

I thank many former colleagues in the Genetic Discovery group at DuPont for many helpful discussions about genome-wide association and eQTL methods. I also especially thank my biologist collaborator and spouse, Ada Ching.

## Author Contributions

**Conceptualization:** Stanley Luck.

**Formal analysis:** Stanley Luck.

**Investigation:** Stanley Luck.

**Methodology:** Stanley Luck.

**Software:** Stanley Luck.

**Writing – original draft:** Stanley Luck.

**Writing – review & editing:** Stanley Luck.

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
