## [Decision Letter · Decision Letter 0]

22 Sep 2021

PONE-D-21-22997The chain rule, measurement error regression and RNA-Seq analysisPLOS ONE

Dear Dr. Luck,

Thank you for submitting your manuscript to PLOS ONE. After careful consideration, we feel that it has merit but does not fully meet PLOS ONE’s publication criteria as it currently stands. Therefore, we invite you to submit a revised version of the manuscript that addresses the points raised during the review process.

Please incorporate the suggestions/comments given by both the reviewers and revise the paper accordingly.

We look forward to receiving your revised manuscript.

Kind regards,

Sriparna Saha, PhD

Academic Editor

PLOS ONE

Journal Requirements:

2. Thank you for providing the following Funding Statement:  

The author, Stanley Luck, is a member of Vector Analytics LLC, which is a science consulting company. Vector Analytics LLC did not have any additional role in the study design, data collection and analysis, decision to publish, or preparation of the manuscript. The specific roles of the author are articulated in the `author contributions' section. The author received no specific funding for this work. 

We note that one or more of the authors is affiliated with the funding organization, indicating the funder may have had some role in the design, data collection, analysis or preparation of your manuscript for publication; in other words, the funder played an indirect role through the participation of the co-authors. 

If the funding organization did not play a role in the study design, data collection and analysis, decision to publish, or preparation of the manuscript and only provided financial support in the form of authors' salaries and/or research materials, please review your statements relating to the author contributions, and ensure you have specifically and accurately indicated the role(s) that these authors had in your study in the Author Contributions section of the online submission form. Please make any necessary amendments directly within this section of the online submission form.  Please also update your Funding Statement to include the following statement: “The funder provided support in the form of salaries for authors [insert relevant initials], but did not have any additional role in the study design, data collection and analysis, decision to publish, or preparation of the manuscript. The specific roles of these authors are articulated in the ‘author contributions’ section.” 

If the funding organization did have an additional role, please state and explain that role within your Funding Statement. 

"Please also provide an updated Competing Interests Statement declaring this commercial affiliation along with any other relevant declarations relating to employment, consultancy, patents, products in development, or marketed products, etc.  

Additional Editor Comments:

As commented by both the reviewers, the paper should be revised based on the suggestions provided.

Reviewers' comments:

Reviewer's Responses to Questions

**Comments to the Author**

1. Is the manuscript technically sound, and do the data support the conclusions?

Reviewer #1: Yes

Reviewer #2: No

2. Has the statistical analysis been performed appropriately and rigorously? 

Reviewer #1: Yes

Reviewer #2: No

3. Have the authors made all data underlying the findings in their manuscript fully available?

Reviewer #1: No

Reviewer #2: Yes

4. Is the manuscript presented in an intelligible fashion and written in standard English?

Reviewer #1: Yes

Reviewer #2: Yes

5. Review Comments to the Author

Reviewer #1: In this paper, the author proposed “The chain rule, measurement error regression and RNA-Seq

analysis”.

The strengths of the paper are that it is well structured, the description of the related work is well done and that results are extensively compared to results of the similar research.

1) Draw a graphical abstract of this work.

2) Justify the novelty of the proposed approach?

3) Proofread the article once again?

4) The experimental validation of the proposed approach is quite confusing justify it.

5) Give some experimental comparison of the proposed approach with existing approach?

Reviewer #2: The manuscript reading looks more like a technical report (heavy on the methodological description). To attract strong interest, it is highly recommended to revise the manuscript to focus on the major objects and findings in a concise manner. I have also some comments which are given below.

1. Title: Consider modifying it to be more specific. The wording of this current title ("The chain rule, measurement error regression and RNA-Seq analysis") is somewhat imprecise.

2. In page no 3 and 4 author mentioned seven novel contributions of his work but they are not clearly established in the whole paper.

3. Recommendation to revise the “data analysis and results” part in details.

4. The discussion part (page no. 23 and 24) does not clearly state the point of view of this paper. The revision is needed for this part.

5. I recommend that the author should revise the Fig. 1, Fig. 5 and Fig. 6 to illustrate the workflow.

6. PLOS authors have the option to publish the peer review history of their article (what does this mean?). If published, this will include your full peer review and any attached files.

Reviewer #1: No

Reviewer #2: No

---

## [Author Response · Author response to Decision Letter 0]

16 Oct 2021

I have provided responses to each point raised by the reviewers in the attached `Response to Reviewers' document.

---

## [Decision Letter · Decision Letter 1]

19 Dec 2021

A parametric framework for multidimensional linear measurement error regression

PONE-D-21-22997R1

Dear Dr. Luck,

We’re pleased to inform you that your manuscript has been judged scientifically suitable for publication and will be formally accepted for publication once it meets all outstanding technical requirements.

Kind regards,

Sriparna Saha, PhD

Academic Editor

PLOS ONE

Additional Editor Comments (optional):

Reviewers' comments:

Reviewer's Responses to Questions

**Comments to the Author**

1. If the authors have adequately addressed your comments raised in a previous round of review and you feel that this manuscript is now acceptable for publication, you may indicate that here to bypass the “Comments to the Author” section, enter your conflict of interest statement in the “Confidential to Editor” section, and submit your "Accept" recommendation.

Reviewer #1: All comments have been addressed

Reviewer #2: All comments have been addressed

2. Is the manuscript technically sound, and do the data support the conclusions?

Reviewer #1: Yes

Reviewer #2: Yes

3. Has the statistical analysis been performed appropriately and rigorously? 

Reviewer #1: Yes

Reviewer #2: Yes

4. Have the authors made all data underlying the findings in their manuscript fully available?

Reviewer #1: Yes

Reviewer #2: Yes

5. Is the manuscript presented in an intelligible fashion and written in standard English?

Reviewer #1: No

Reviewer #2: Yes

6. Review Comments to the Author

Reviewer #1: All my comments are addressed by the authors clearly and explanations given by authors is impressive.

Reviewer #2: Author tried to make the changes according to the instructions. It looks better than previous though the introduction portion is not up-to the mark.

7. PLOS authors have the option to publish the peer review history of their article (what does this mean?). If published, this will include your full peer review and any attached files.

Reviewer #1: No

Reviewer #2: No

---

## [Editor Report · Acceptance letter]

22 Dec 2021

PONE-D-21-22997R1 

A parametric framework for multidimensional linear measurement error regression 

Dear Dr. Luck:

I'm pleased to inform you that your manuscript has been deemed suitable for publication in PLOS ONE. Congratulations! Your manuscript is now with our production department. 

Kind regards, 

on behalf of

Dr. Sriparna Saha 

Academic Editor

PLOS ONE